# Balancing the Poles of the Seesaw: The Parallel Paths of Eckhart and Hindu Vedānta toward Oneness with God/Brahman

**Jianye Liu** [1] and **Zhicheng Wang** [2,*]

1   School of Philosophy, Zhejiang University, Hangzhou 310058, China; liujianye@zju.edu.cn
2   Institute for Marxist Religious Studies in New Era, Hangzhou City University, Hangzhou 310015, China
*   Correspondence: wangzhicheng@hzcu.edu.cn

**Abstract:** The ultimate aim of both Eckhart's philosophy and Vedānta philosophy is to attain oneness with God/Brahman. Nevertheless, their different philosophical starting points and the conflict between the sublime ideal of the theory and reality means that their philosophies present a structural symmetry. They both have to face two dilemmas: "How can we claim that humans are already one with God?" and "Why is it that humans are not already one with God?". Eckhart's inherited tradition emphasizes the distinction between humans and God, while the Vedānta philosophical tradition emphasizes that "*I am Brahman*". Each of them starts from one pole of the seesaw of the dilemma and encounters the other's issue at the other pole. Eventually, they converge at the point of balance, with unity with God/Brahman realized in all human activities. Here, this worldly life becomes significant, all human work expresses the Divinity, and the importance of God is replaced by an impersonal Divinity that combines being and nothingness.

**Keywords:** Vedānta; Brahman; Śaṅkara; Eckhart; theology; comparative philosophy

## 1. Introduction

There are similarities between Meister Eckhart's philosophy and Vedānta philosophy, as addressed in numerous existing studies. As Milne (1993) suggests, one of the primary reasons for comparing Eckhart with Advaita Vedānta is that non-dualism is not only present in Eastern religious philosophy but also embedded within Western religious philosophy. Non-dualism lies at the very heart of Christianity, in the person of Christ. Although man and God are different, Christ is both God and man. The Church has long struggled to find appropriate ways to articulate the true nature of Christ. Eckhart's philosophy embodies a typical non-dualistic thought, wherein the humanity and Divinity of Christ are not separated. Milne's research primarily focuses on non-duality, asserting that in both Shankara and the works of Eckhart, non-duality is fundamental to the deepest level of religious consciousness. In contrast, we concentrate on the dynamic evolution of non-duality within the philosophies of Eckhart and Vedānta. Evola (1960) notes that Vedānta, Eckhart, and Schelling have the same conception of the Supreme Unity, the Eternal One. All of them consistently oppose theism as the supreme frame of reference. In the case of Eckhart, he emphasized the distinction between God and Divinity, and the "noble soul" must detach itself, break through God, and move forward into the "desert" of the Divinity. Evola considers Vedānta as a metaphysical, esoteric tradition and as transpersonal. We basically agree with him, but the development of the neo-Vedānta has made Vedānta not merely "superpersonal", and it is in this change that we see the structure of symmetry and the eventual convergence between Vedānta and Eckhart's mystical philosophy. Shah-Kazemi (1997) also carefully examines the philosophies of Eckhart and Śaṅkara regarding the two basic philosophical positions of transcendence and immanence. However, these studies primarily compare the similarities in the intricate details of their theories—although these are perhaps the most important details. None of them gives importance to the neo-Vedānta,

whereas the current article gives importance to a kind of symmetrical structure with Eckhart's philosophy that emerges throughout the whole development of Vedānta.

This paper argues that there are structural similarities between Eckhart's philosophy and Vedānta philosophy, beyond merely the generally mentioned similarity of the unity or identity of man and God (Brahman). Both philosophies contain contradictory elements. For example, Eckhart, on the one hand, emphasizes the renunciation of our body and work; on the other hand, he stresses the unity of our physical body with the body of God, this unity with God in secular affairs, and the realization that all things exist within God and that everything in the world is God. On the more academic side, as Siwen (2023) suggests, Eckhart is influenced by the Thomistic tradition on the one hand, which understands God (Gott) as Being and First Cause, and, on the other hand, by neo-Platonism and mysticism, which emphasizes a more fundamental "Divinity" (Gottheit, Godhead) as nothing, as well as the unground, which is distinct from God. God and Divinity are neither identifiable nor distinguishable in this tension, as the absolute one and nothing. Vedānta philosophy similarly presents two contradictory ideas: one considers the world as impermanent and full of sufferings and thereby promotes the renunciation of the world to attain liberation, while the other views all as the Absolute (Brahman), with the entire world being the Absolute, and thereby advocates unity with the Brahman in all activities, including work, life, and even breathing. Similar to Eckhart's distinction between God and Divinity, there is a distinction in Indian philosophy between Brahmā, one of the major personality gods of Hinduism, and Brahman, the divinity that is immanent in and transcends all; however, Vedānta is not concerned with Brahmā. The apparent contradiction above suggests that both philosophies are not merely static theories; instead, each represents a trajectory of theoretical development, thus containing theoretical tension. Therefore, it is reasonable to compare Eckhart's philosophy with the long tradition of Vedānta philosophy.

We know that both philosophies emphasize the identity of humans and God/Brahman, and they both ultimately stress the importance of this worldly life. However, their starting points and theoretical trajectories differ. It is clear that any philosophy rejecting transcendence and emphasizing identity and immanence has to face a dilemma due to the limited horizon of this worldly life: "How can we claim that humans (living in this finite world) are already one with God?" (issue A) and "Why is it that humans are not already one with God (so we are still living in this finite world)?" (issue B). These two issues are like two poles of a seesaw, with different presuppositions. Eckhart's philosophy and Vedānta philosophy originated at one pole of the seesaw, and as they developed, they slid to the other pole of the seesaw and eventually balanced themselves. Eckhart's philosophy constantly teaches people how to seek God[1]; in a theory that looks to God from the perspective of this worldly life, his presupposition is: "How can we claim that humans are already one with God?" Vedānta philosophy begins with the Upanishadic statement that "*I am Brahman*", thereby facing the experiential world filled with sufferings; it starts with the question: "Why is it that humans are not already one with God?"

To better present the symmetrical structure of this seesaw, this paper divides each of these two philosophies into three stages. However, note that this distinction of stages does not begin with the latter stage canceling out and replacing the former; they always coexist in tension. In Eckhart's philosophy, these three stages are: (1) the human–God difference and the difference between God and the Divinity; (2) becoming one with God through self-emptying and the Divinity in the depths of God; (3) becoming one with God in Being and the God–Divinity relationship is without identity or distinction. For Eckhart, there are four stages to the union between the soul and God: dissimilarity, similarity, identity, and breakthrough (Schürmann 2023). Here, the second and third stages merged into the middle stage, according to our way of division. From stages 1 to 2 and 3, Eckhart deals first with issue B and then issue A. Yet, since Vedānta philosophy is derived from the *Upaniṣads*, which emphasize the immanence of ritual, it begins with an emphasis on "*I am Brahman*", and the significance of Brahmā is eliminated at the outset. Corresponding to this, approach, Eckhart's philosophy deconstructs the significance of God and transfers it

to an impersonal Divinity in stage 2. In Vedānta philosophy, the three stages are: (1) "*I am Brahman*"; (2) *Māyā* is the obstacle to our unity with Brahman; (3) everything is Brahman. In stage 1, Vedānta philosophy is faced with issue A. For this reason, Śaṅkara regards the ego, representing this world, as being false, which brings Vedānta philosophy to stage 2 and encounters issue B. In this article, we depict Eckhart's philosophy in Section 2 and Vedānta philosophy in Section 3 and show how both move toward balance in Section 4.

## 2. Eckhart: From Human–God Difference to Oneness with God

Eckhart is a devout lover of God, adhering to the creed that God is not a creature but is purely creating. It is in this sense that a series of Eckhart's theological theories are integrated. Based on it, his primary approach is what is termed speculative theology. The term "speculative" here is not dialectic but makes philosophy a method: philosophy is the love of wisdom (philo-sophia) and not the possession of wisdom. Thus, speculative theology means to love God in both a religious way and in the way of loving wisdom. This is the profound love we read of in Eckhart's philosophy, where he trusts God unconditionally and requires us to do the same. It was based on this love that he developed the philosophy of detachment.

The fundamental intuition and standpoint of Eckhart's philosophy lies in the belief that God is not a creature like human beings but instead is creating and is even Love itself. It is in this sense that Eckhart distinguishes between God (Gott) and Divinity (Gottheit, Godhead), in which divinity "is superior to being and is unnameable, naked simplicity" (Evola 1960), and God remains personified. Eckhart (1957, p. 278) argues that "there is nothing prior to being because that which confers being creates and is a creator. To create is to give being out of nothing. It is a fact that all things have being from being itself, just as all things are white from whiteness itself. Therefore, if being is different from God, the creator will be something other than God". In this sense, it can be said that being is equivalent to God, who is the Creator, and that the creating itself is nothing, that is, Divinity, and man is the creature. Although we liken the divine to creating itself, it is, like Aristotle's "entelecheia", the immovable driver. In Eckhart, the "nothing" of Divinity is the ground without ground that is behind all creating. Although this doctrinal distinction exists, for the sake of the sermon, Eckhart's use of the word "God" often carries the dual identity of creating and creator, nothing and being—and this is easy to understand. On the one hand, God and man are equal, while creating is total silence; it is nothing, it does not even act. On the other hand, in the case of the creature, the creator is often regarded as a synonym for creating itself. Therefore, "God becomes and unbecomes … God works, the Godhead does no work: there is nothing for it to do, there is no activity in it. It never peeped at any work" (Eckhart 2009, pp. 293–94). Similarly, Eckhart emphasizes that "being is God" (Eckhart 1957, p. 278), rather than "God is being", because it represents both "being" and "nothing".

Since the word "God" has two meanings, the relationship between humans and God is always influenced by the relationship between God and Divinity. Although we always speak of a "human–God relationship", "God" has both transcendent and immanent, being and nothingness dimensions. The complex relationship between God and Divinity is expressed in the phrase "the God beyond God" (cf. McGinn 1981). Since the word "God" has two meanings, the relationship between humans and God is always influenced by the relationship between God and Divinity. Although we always speak of a "human–God relationship", "God" has dimensions that are both transcendent and immanent, being and nothingness. The complex relationship between God and Divinity is expressed in the phrase "the God beyond God" (cf. McGinn 1981). The difference between man and God that is emphasized in this section is fundamentally the difference between God and Divinity, and this is issue B with which Eckhart is faced; for this reason, the solution seems to be that the depths of God are Divinity, and, thus, the solution to the union of man with God is embodied in the separation of being and self-emptying, which brings Eckhart back to issue A. This specialness of God is also an issue that must be broken through; only then can

true detachment and freedom be attained. In Section 4, we will talk about the way Eckhart moves from the pole of the seesaw with issue A to the point of balance.

A contradiction seems to arise when we contemplate how to approach God—if humans can pursue God on their own initiative, does this imply that humans and God are separate and independent entities? Therefore, Eckhart (2009, p. 414) asserts that the essence of creatures is nothingness and God brought them into existence. If man wants to pursue God, he has to make God pursue him by bearing his inherent nothingness, which means rejecting the body and the works of the ego. However, since everything is God's creation from nothingness, the body and the works of the ego are no exception. Therefore, Miles (2005, p. 194) said of Eckhart that "he sometimes said that body, intellect, and works are to be totally rejected; in other passages, all of these seem to be affirmed". Eckhart's line of thought always presents its various logical stages simultaneously, especially in his sermons. His ideas manifest as a dynamic developmental process, preserving the tension between the love of God and the love of wisdom. In contrast to the gradual transitions in Vedānta philosophy over an extended historical period, the tension in Eckhart's philosophy is not a temporal transition but rather an intrinsic tension within his philosophy. In today's terms, his thinking contains six closely linked and even mutually inclusive logical segments: God is not created, God is different from humans (creatures), humans approach God through detachment/God comes to humans through detachment, oneness with God, the omnipresence or immanence of God, and how humans should live based on these.

In his view of body and soul opposition, Eckhart was influenced by the theological tradition—at least by the Neoplatonism and Augustinian approaches that were prevalent in his time. In similar views, the body is seen as a burden for the soul, hindering the path to God. Even if Eckhart did not fully accept the notion of the body as a burden, he nevertheless acknowledged the existence of obstructions and impediments to a certain extent; he "…cuts away the chips that had hidden and concealed the image: he gives nothing to the wood but takes from it, cutting away the overlay and removing the dross" (Eckhart 2009, p. 560).

In Augustine's view, to avoid obstruction, one should seek truth inwardly. This is similar to the transition found in Indian philosophy from external sacrifice to internal sacrifice. However, the problem lies in the fact that the internal still exists relative to the external, and claiming to seek God within humans almost implies that humans are some independent and sufficient entity. In Eckhart's case, on the other hand, the situation is quite different. If God is purely creating and all creatures lie in God, then humans as creatures cannot inherently possess themselves. Humans are utter nothingness, where "all creatures are pure nothing. I do not say they are a trifle or they are anything: they are pure nothing … If God turned away for an instant from all creatures, they would perish" (Eckhart 2009, p. 226). God is who He is without any specific form. In his different view of creatures and of God, Eckhart does not focus on the ordinariness, materiality, and problem of evil in humans as theologians before him did. He does not belittle humans relative to the supreme God but instead elevates God even higher, placing Him thoroughly on the level of non-creature. He even says, "He is one and indivisible, without mode or properties: in that sense, He is neither Father, Son, nor Holy Ghost, and yet is a Something which is neither this nor that" (Eckhart 2009, p. 81). All creatures are created out of nothingness. In Sermon 19, Eckhart (2009, pp. 137–38) elaborates on the four meanings of the phrase: "Paul rose from the ground and with open eyes saw nothing", creatively interpreting the nothingness of everything as God's seeing.

Humans are creatures, which means that humans do not possess themselves in essence; they are utter nothingness. Nevertheless, along the path of nothingness, humans can also return to God. This is because the way in which humans seek God differs from the way that they seek other creatures. The pursuit of God by humans is to allow God to approach them. All of this is part of Eckhart's theory of self-emptying. This absolute self-emptying can "force" God to fill the vacuum in the soul (Vinzent 2011, p. 222). It is important to note

that the theory of self-emptying is a methodology that always serves the pursuit of God. In Blakney's translation, he quotes the following passage:

> *Someone complained to Meister Eckhart that no one could understand his sermons. Whereupon he said: "To understand my preaching, five things are needed. The hearer must have conquered strife; he must be contemplating his highest good; he must be satisfied to do God's bidding; he must be a beginner among beginners; and, denying himself, he must be so a master of himself as to be incapable of anger".* (Eckhart 1957, p. 93)

The most important of these five things seems to be the necessity to "be a beginner among beginners", an expression synonymous with "emptying oneself", i.e., accepting one's ignorance rather than asserting knowledge. This is the practice of self-emptying, which is precisely the way to reach God. Denying oneself and avoiding anger are means to this end. However, one must also "contemplate his highest good", for everything that remains after the ego is discarded comes from God and is the good of God. This good represents how a connection with God is established, and to "be satisfied to do God's bidding" is both within this means and is an end in itself. Ultimately, one conquers the strife between ego and God and remains a beginner who follows God's bidding. As Eckhart (2009, p. 518) often expressed, "For the more we possess of things, the less we possess of Him, and the less love we have of things, the more we have of Him with all that He can do".

The obstacle to the soul is the soul itself. Even for the very devout, if their pious souls always care about pursuing God for themselves, they will never be able to touch God, and even the very devout will never be able to reach God if their pious souls are always concerned with pursuing God for their own benefit (Vinzent 2011, pp. 60–61). In order to pursue God, one has to accept all of God. It is a delusion to think that one can accept God's grace on the one hand while still retaining the ego on the other. "God must give me Himself for my own as He is His own, or I shall get nothing, and nothing will be to my taste. Whoever shall thus receive Him outright must have wholly renounced himself" (Eckhart 2009, pp. 226–27). The creature is characterized by nothingness, which is both the drawback that makes the creature different from God and the key to the creature's ability to be one with God. At this stage of logic, therefore, Eckhart criticizes works on the ego level. The works he criticizes are those performed for oneself rather than for God, treating works as ends rather than means.

Self-emptying represents a journey away from the self toward complete de-objectification. Even an objectified concept of God is no longer pursued in this state. It is only then that God truly arrives. As Eckhart said:

> *If a man turns away from self and from all created things, then … you will attain to oneness and blessedness in your soul's spark,9 which time and place never touched. This spark is opposed to all creatures: it wants nothing but God, naked, just as He is. It is not satisfied with the Father or the Son or the Holy Ghost, or all three Persons so far as they preserve their several properties … it seeks to know whence this being comes, it wants to get into its simple ground, into the silent desert into which no distinction ever peeped, of Father, Son or Holy Ghost … this ground is an impartible stillness, motionless in itself, and by this immobility all things are moved, and all those receive life that live of themselves, being endowed with reason.* (Eckhart 2009, pp. 310–11)

This suggests that in order to deal with issue B, Eckhart believes that we need to turn away from all creatures and all beings. Furthermore, the God that man attains in this state is not the Father or the Son or the Holy Ghost, but the Divinity. In this case, the Divinity is in the depths of God[2], and the two are different. "God and Godhead are as different as heaven and earth. I say further: the inner and the outer man are as different as heaven and earth" (Eckhart 2009, p. 293). Reaching oneness with God through complete self-emptying seems too difficult and idealistic to be possible. Can living beings actually reach God? Or is it true that only by dying can one reach nirvana, as the primitive Buddhists believed? This means that Eckhart has to deal with issue A.

In oneness with God (Divinity), "we should be turned into Him and become fully united with Him, so that His own becomes ours, and ours all becomes His: our heart and His one heart and our body and His one body" (Eckhart 2009, p. 509). We will discuss this in detail in Section 4, while here we would like to briefly mention that not only the soul but also the body, which is traditionally regarded as impure and burdensome, can be one with God. On this point, he differs from previous theologians. He not only talks about how all things show the greatness of the Creator but also emphasizes the importance of creatures for God: "For before there were creatures, God was not 'God': He was That which He was. But when creatures came into existence and received their created being, then God was not 'God' in Himself, He was 'God' in creatures" (Eckhart 2009, p. 421). God is born in humans, and humans are born in God (Eckhart 2009, p. 92). Note that God does not depend on creatures and is not only realized in creatures—this is perhaps the most significant difference between Eckhart's and Spinoza's philosophy; God (Divinity) is still beyond the creature. Although Eckhart did not accept the overflow theory, he did not wholly move towards a Spinoza-style philosophy. Vedānta philosophy precisely echoes this point with him—everything is Brahman, but Brahman is not solely all these things.

### 3. Vedānta: From "*I Am Brahman*" to "*Māyā*"

Vedānta is a significant branch of traditional Indian philosophy that remains active both in India and worldwide today. The literal meaning of Vedānta is the end of the Vedas. However, the end of the Vedas does not mean the end of or disappearance of the Vedas but is, rather, a summation and development. The Vedic texts can be broadly categorized into four groups—the Samhitas, Brāhmaṇas, Āraṇyakas, and Upaniṣads. The Samhitas consist mainly of hymns dedicated to the gods; the Brāhmaṇas explain the complex rituals based on the Vedas and interpret the mystical content of these ceremonies; the Āraṇyakas are books on hermitage in the forest; and the Upaniṣads are philosophical contemplations of the real, the source of the universe, and the nature of man. These classics "may be roughly characterized as the successive utterances of poets, priests, and philosophers" (Hume 1921, p. 5). According to Georg Feuerstein (2008, pp. 240–93), progressing from the Samhitas to the Brāhmaṇas, the standardization of Vedic rituals was completed. The Brāhmaṇas are texts used for Brahmanic rituals, describing their origins, meanings, and benefits. Later, the Āraṇyakas refer to the knowledge that was imparted in the forest. Due to the lack of resources for performing rituals in the forest, these ritual activities gradually shifted from physical practice to imaginative contemplation. By the time of the Upaniṣads, some practitioners transitioned from conducting sacrifices in meditation to purely philosophical contemplation. For instance, as is written in the Śvetāśvatara Upaniṣad:

> *The Brahma-students say: Is Brahman the cause? Whence are we born? Whereby do we live, and whither do we go? O ye who know Brahman, tell us at whose command we abide, whether in pain or in pleasure. Should time, or nature, or necessity, or chance, or the elements be considered as the cause, or he who is called the Purusha?* (Muller 1884, pp. 231–32)

The characteristic of this evolution from Brāhmaṇa to Upaniṣad is that specific ritual ceremonies that are performed for the sake of gaining benefits turn gradually into imagining the process of sacrifices in the mind, then turn into contemplating the symbols and metaphors of ritual ceremonies, and finally turn entirely into philosophical thinking. In other words, it is a transition from an external, formalized natural religion to a spiritual religion with true inwardness and transcendence. Radhakrishnan (1953, p. 8) describes the importance of the Upaniṣads for religious rituals as follows: "The Upaniṣads, which base their affirmations on spiritual experience, are invaluable for us as the traditional props of faith, the infallible scripture, miracle and prophecy are no longer available".

The core concept of the Upaniṣads is "aham brahmāsmi" (I am Brahman). The Bṛhadāraṇyaka Upaniṣad (Mādhavānanda 1950, p. 92) states, "In the beginning, this (universe) was but the self (Viraj) of a human form. He reflected and found nothing else but himself. He first uttered, 'I am he.' Therefore he was called Aham (I)". This means that the essence

or true self of man is Brahman. In any religion, especially a theistic one, it is considered appalling to claim that the essence of man is the same as Brahman/God.[3] Although the Upaniṣads belong to the orthodox Vedic religion, we can imagine how the Vedic sacrificial clergy would have resented the claim that "I am Brahman", which suggests that to achieve liberation, one need not rely on the external, cumbersome Vedic sacrificial rituals or personal gods. Instead, liberation can be attained through the inward contemplation of one's own intrinsic nature. We must note that Hinduism encompasses both transcendent personal deities, as depicted in the Vedic scriptures, and a metaphysical impersonal deity, as represented by Vedānta. Therefore, a brief explanation of them is necessary. Brahmā, Vishnu, and Shiva are primary deities; they all have temples, idols, incarnations, legends, and many followers. Strictly speaking, they are not the purest existences in philosophy. Furthermore, the Upanishads say, "I am Brahman", not "I am Brahma", and also state, "Thou art Brahma, thou art Vishnu, thou art Rudra (Shiva), thou art Agni, Varuna, Vayu, Indra, thou art All" (Hume 1921, pp. 422–24). Without a doubt, India's theistic tradition is extensive and sophisticated, with a broad spectrum of tantric practices. Nonetheless, the Vedānta, which this paper discusses, typically gravitates more toward metaphysical discourse and the investigation of an impersonal deity.

From the Samhita to the Upaniṣads, the Vedic religion undergoes a complete transition from external worship to internal spiritual worship. G.C. Pande (1999, p. 603) tells us that the statement "I am Brahman" marks the conclusion of India's lengthy inquiry into absolute existence. However, this conclusion is also the starting point for all subsequent discussions within the Vedānta schools. The Brahman found in the Upaniṣads is the supreme God, the supreme existence, and the supreme Self. It transforms Itself to generate the world and the true Self (ātman) of humans. It not only penetrates all things as an inner essence but also includes all things within itself as an indivisible One. The essence of the world is Brahman, and everything is Brahman. The Upaniṣads have dominated the philosophy, religion, and life of Indians for three thousand years. As Sarvepalli Radhakrishnan (1953, p. 17) says, "Every subsequent religious movement has had to show itself to be in accord with their philosophical statements … They have survived many changes, religious and secular, and helped many generations of men to formulate their views on the chief problems of life and existence".

If everything is Brahman, and our essence is Brahman ("I am Brahman"), why is it that both I and this world are limited and imperfect? This is a question that Vedānta philosophers are inevitably confronted with. Dr. B.N.K. Sharma (1986, p. 12) believes that the key to solving this problem lies in managing the relationship between "the thinking self, a world of external realities, and indications of an Infinite Power rising above them". Different answers to this problem have formed different schools of Vedānta. The three main schools are Advaita Vedānta, as represented by Gaudapāda and Śaṅkara, the Viśiṣṭā Advaita of Rāmānuja, and the Dvaita of Madhva. Before explaining the first school, let us briefly introduce the last two. The term Viśiṣṭā Advaita means "qualified non-duality". It asserts that Brahman (the ultimate reality), the individual self, and the phenomenal world composed of matter all truly exist. The individual self and the world are expressions of Brahman's attributes, and the reality of the individual self and the world limits Brahman. It is clear that the problem with Viśiṣṭā Advaita is that, as the origin of the world, Brahman has attributes and it is limited by these attributes, which negates the absoluteness of Brahman. The term Dvaita means "dualism". Madhva attests that there is no absolute inseparability between God and the world, as well as between God and the self. He opposes the view that considers the world and the self as the essential nature of God. According to Jeaneane D. Fowler (2002, p. 342), the philosophical school holds a distinct dualistic perspective on reality, asserting that the temporal and the divine are separate.

Gaudapāda is the founder of Advaita Vedānta, and Māyā represents his theory for dealing with the relationship between the self, the world, and God (Brahman). He believes that both subjective and objective things are illusory. What is Māyā? According to him, "That which is non-existent at the beginning and in the end is necessarily so (non-existent)

in the middle" (Nikhilānand 1949, p. 97). Swāmī Nikhilānand (1949, pp. xii) interprets that according to Gaudapād, illusory objects do not exist at the beginning and the end; they are all non-eternal, non-real, and, furthermore, are ultimately subject to decay and death. However, Māyā is not the nihilism of Western philosophy, for behind Māyā there is Brahman, the ultimate reality. Gaudapād believes that everything that people experience is nothing but the non-dual Brahman. The perception of duality arises due to our ignorance, as we are unaware of the true nature of the Absolute, which is the non-dual Brahman. In short, he asserts that Brahman and the essence of man (the self) are one, and the world is only an unreal illusion of Brahman.

Śaṅkara is the most crucial philosopher of Advaita Vedānta. He inherits the ideas of Gaudapāda and makes significant developments based on them. Similarly, Śaṅkara also believes that Brahman is the origin of the world. However, he develops two concepts: Brahman with qualities (Saguṇa Brahman) and Brahman without qualities (Nirguṇa Brahman). According to Monier-Williams (2022), Guṇa means "quality, peculiarity, attribute, or property". Saguṇa Brahman refers to Brahman with qualities, while Nirguṇa Brahman refers to Brahman without qualities. Therefore, regarding the epistemology, there is a lower level of knowledge (Aparā Vidyā), which corresponds to Saguṇa Brahman, and supreme knowledge (Parā Vidyā), which corresponds to Nirguṇa Brahman. As Saguṇa Brahman, it is the source of the universe, such as God, that ordinary people can understand and imagine. As Nirguṇa Brahman, it transcends the entire experiential world and the real world. It is the ultimate reality and eternal existence, but ordinary people cannot understand the existence of the supreme Brahman. Only through the continuous search for knowledge of Brahman can one have insight into the Nirguṇa Brahman. In Gaudapāda's view, human experience is only a creation of Brahman, so the world is only an illusion (Māyā). But, in Śaṅkara's view, Brahman has a power called Māyā. Jeaneane D. Fowler (2002, p. 246) explains that according to Śaṅkara, we live in a world of illusory phenomena to which we give names and to which we cling in the belief that they are real. Māyā projects and produces the world that people experience, but, at the same time, it also veils us, preventing us from recognizing the Brahman behind the experiential world.

We can see that the philosophical starting points of Eckhart and Vedānta are different. Eckhart starts from the basis of the difference between man and God (the opposition of flesh and spirit) and explains why man is one with God; Vedānta starts with "I am Brahman" and explains why I and the world do not manifest as perfect Brahman. We can regard this as a diagnosis: firstly, the essence of a person is Brahman, so the healthy (ideal) state of a person should mirror that of Brahman—being eternal and joyful. However, in reality, people and the external world are constantly subject to change and are ultimately destined to decay and die. Vedānta philosophy believes that the cause of disease is human ignorance, i.e., being veiled by Māyā, not seeing Brahman behind the experiential world, and not realizing that the experiential world is also unreal. The method of curing this disease is "vichāra", which means discrimination. It is the faculty of discrimination that distinguishes the real, Brahman, from the unreal. That is, we must shed the things in this world that belong to us. As Śaṅkara (Mādhavānanda 1921, p. 33) says, this gross body is to be deprecated, for it consists of the skin, flesh, blood, arteries and veins, fat, marrow, and bones, and is full of other offensive things. However, we soon discover the paradox of it, namely, that although "I am Brahman", Brahman is so different from the self as an individual, "like the sun and a glowworm, the king and a servant, the ocean and a well, or Mount Meru and an atom" (Mādhavānanda 1921, p. 108). Brahman is within everything and yet transcends everything, "that beyond which there is nothing; which shines even above Māya, which again is superior to its effect, the universe; the inmost Self of all" (Mādhavānanda 1921, p. 119). Within the principle that everything is Brahman, how can we reconcile the contradiction between real and unreal, immanence and transcendence, One and diversity? As can be seen, although Eckhart and Vedānta start from different poles of the seesaw, they both want to find a theoretical balance. The issue with which the philosophy of Vedānta is concerned at the present stage is this: if man and God are the same,

why are humans not already one with God (so that we are still living in this finite world)? Eckhart approaches this issue from a different perspective and he deals with it in the next theoretical stage because of his different starting points. Nevertheless, they will eventually realize that seeking balance without going to the opposite end of the seesaw means acknowledging that there is no difference between humans and God. It requires removing human specificity and ego for humans to act as though "all is Brahman/God."

### 4. Balance of Theories and the Revival of Works

If we only emphasize oneness with God and "I am Brahman" by excluding something, we still treat God/Brahman as the supreme object, failing to make it wholly univocal and immanent. It only reveals the union of humans and God based on the difference between God and the Divinity. Here, the detachment is revealed in a more fundamental way. For Eckhart and in Vedānta philosophy, the dilemmas between "Why is it that humans are not already one with God?", and "How can we claim that humans are already one with God?" have not been properly resolved. In the previous discussion, Eckhart explained how humans are already one with God, while Vedānta philosophy explained why humans have not yet actualized oneness with God. Each has moved to the other pole of the seesaw, and, thus, both need to return to find the balance. The limitation of the theory at this stage is that, whether considering Eckhart's or Vedānta philosophy, they both pay attention to God while overlooking everything in this worldly life. The balance of the seesaw, however, lies in the importance of all creatures being one with God and the importance of work in this world.

As pure creating and pure love, God (as Divinity) can be loved by humans in the same way that God loves humans, even though humans lack the ability to create. To love is to give completely, without any reservations for oneself—this inherently implies detachment. "But divine love takes us into itself, and we are one with it … thus all creatures are maintained in existence by love, which is God" ([Eckhart 2009](), p. 63). On the one hand, all creatures exist because of love, and love is synonymous with God; on the other hand, we, the creatures of God's love, pursue God, precisely through love. God is love, and we aspire to God through love; love can only aspire to God, and only love can aspire to God.[4] Through self-emptying, love, and detachment, we gradually journey toward oneness with God. Detachment is the complete manifestation of self-emptying, and love also means giving entirely without reservation. Whether self-emptying or love, they are all subsumed under the fundamental yet complex concept of detachment[5]. Eckhart believes:

> *This immovable detachment brings a man into the greatest likeness to God. For the reason why God is God is because of His immovable detachment, and from this detachment He has His purity, His simplicity, and His immutability. Therefore, if a man is to be like God, as far as a creature can have likeness with God, this must come from detachment. This draws a man into purity, and from purity into simplicity, and from simplicity into immutability, and these things make a likeness between God and that man; and this likeness must occur through grace, for grace draws a man away from all temporal things and purges him of all that is transient.* ([Eckhart 2009](), p. 569)

Detachment, on the one hand, refers to humans' detachment (completely giving of oneself and moving toward God) and, on the other hand, God's detachment (completely giving of Himself and moving toward people). In the absolute detachment of God, we see the divinity, and detachment is a manifestation of love for God, which means to love God with philosophical love. Furthermore, detachment goes beyond loving God. Even the subject of loving God is effaced, leaving only pure love, the Divinity. "You should love Him as He is: a non-God, a non-spirit, a non-person, a non-image; rather, as He is a sheer pure limpid One, detached from all duality" ([Eckhart 2009](), p. 465). This also means that with detachment, one no longer needs to grovel or even exhibit humility before God— humility still retains the self or ego, while detachment is in oneself but means forgetting oneself.[6] "That sounds strange, that man can become God in love, but so it is true in the eternal truth,

and our Lord Jesus Christ possesses it" (Eckhart 2009, p. 105). If self-emptying is analogous to humility, starting from the basis of self or ego and continuously lowering oneself permanently preserves the basis of self. While detachment no longer demands an ego, it no longer generates any demands and desires from the ego, thus entering into oneness with God with complete simplicity. Humility is obedience to transcendence and detachment to immanence. The grace that is given by God is nothing less than God Himself.

For Eckhart, the concept of detachment does not merely remain at the level of self-emptying or should be treated solely as a methodological approach. It must evolve into an ontological perspective to radicalize the theory. Notions assuming that there are hindrances and absolute evil are not radical enough. For instance, when discussing the abandonment of the ego, Eckhart cited the following examples: "It is in the darkness that one finds this light" (Eckhart 2009, p. 410) and "What is a pure heart? That is pure which is separated and parted from all creatures, for all creatures produce impurity because they are nothing, and nothing is a lack and tarnishes the soul" (Eckhart 2009, p. 105). This precisely moves the discourse from the issue of "How can we claim that humans are already one with God?" to its antithesis, "Why is it that humans have not already actualized their oneness with God?", i.e., why are we still in darkness and stained if all creatures are of God? The issue is faced at the starting point of Vedānta philosophy, as described in the previous section. Vedānta philosophy has similarly slipped from one pole of the seesaw to the other, facing up to Eckhart's philosophy at the starting point.

The body represents Eckhart's theoretical breakthrough. In previous stages of the theory, the body and soul were often opposed (Eckhart 2009, p. 59), but now, "my body is more in my soul than my soul is in my body, but both body and soul are more in God than they are in themselves" (Eckhart 2009, p. 334). In Eckhart's view, both the body and the soul are creatures that have a direct relationship with Divinity first, and with each other second. Eckhart negates the Platonic approach of hierarchically ordering God, soul, and body. He denies any intermediaries to God since any intermediary, by analogy, sees God as created.

In this unity with God, the body and soul fit together. As Eckhart notes, this is precisely the way to love each other through the love of God (see Eckhart 2009, p. 101, for more detail). If such love is genuinely equal, it should regard all God's creatures equally, including the body and soul. If the soul (excluding the ego) is where God manifests, then the body must also be where God resides. The saying "He was 'God' in creatures" (Eckhart 2009, p. 421) already indicates that God (Divinity), by creating itself, omnipotently unifies all creatures. No creature does not exist and act in and for the sake of Divinity. While the Divinity does not become something, it remains quiet and is nothing. Only then can all beings come equally from the Divine. The relationship of God and Divinity is without identity or distinction in this stage. This is why Eckhart said that "God becomes and unbecomes" (Eckhart 2009, p. 293) and that "God works, the Godhead does no work" (Eckhart 2009, p. 294).

At this point, the personified God is dissolved in man's relationship with Divinity (although it is still called God). "When I return to God, if I do not remain there, my breakthrough will be far nobler than my outflowing" (Eckhart 2009, p. 294). Just as in Śaṅkara, Brahman appears in "I", with Māyā removed. We have to give up everything, even detachment from God, for Divinity is undoubtedly both univocal and absolutely quiet, a synthesis of being and nothingness. Pure nothing means absolute detachment. At this stage, the so-called appearances, obscurations, and stains do not accurately describe the facts, much like the development of Vedānta philosophy, which also began to emphasize that there is no need to leave the world for liberation. The truth is that once we become one with God/Brahman (which means Divinity from now on), we will acquire a univocal vision, seeing everything in the world as God/Brahman. Eckhart said:

> I am often asked if a man can reach the point where he is no longer hindered by time, multiplicity, or matter. Assuredly! Once this birth has really occurred, no creatures can hinder you; instead, they will all direct you to God and this birth … In fact, what used

*to be a hindrance now helps you most. Your face is so fully turned toward this birth that, no matter what you see or hear, you can get nothing but this birth from all things. All things become simply God to you, for in all things you notice only God, just as a man who stares long at the sun sees the sun in whatever he afterward looks at. If this is lacking, this looking for and seeking God in all and sundry, then you lack this birth.* (Eckhart 2009, p. 59)

Eckhart lived at a time when most mystics and theologians believed that one had to renounce the things of this world in order to reach God (Miles 2005, p. 198). Eckhart also shared this view, considering it part of the theoretical stage of self-emptying. However, at this stage, he now recognizes that everything points to God, and what were once considered obstructions and stains have turned into signposts on the way to seeking the Divinity that is beyond time and space, with the body and its works being returned together. Similarly, Vedānta philosophy has also undergone a process of rejection and reclaiming works, shifting from external ritual to internal worship and then to contemporary works. When all paths are seen as leading to God, how should one act? Eckhart argued that if a person does not yet have the absence of the need for action and forces himself to empty all concepts and refrain from action, then he should still act (work). We must align our external actions with our inner intentions, being driven by a sense of destiny rather than working aimlessly. In this way, "one is co-operating with God" (Eckhart 2009, p. 517), which means that everything comes from the Divinity who does no work directly. Nothing is evil; the path to God originally had no obstacles, and people reached God through various works. These works were initially good but became obstacles when the people's desires were for the works and actions rather than for God, which is the purpose of works (Vinzent 2011, p. 155). For example, a title that is given according to ability is good, but its pursuit will lead us astray if a title itself becomes the object of that pursuit.

Eckhart sometimes emphasized that we are to seek God in many ways and also sometimes singly: "God has not bound man's salvation to any special mode … We should have more regard to other people's way, when they have true devotion, and not scorn anybody's way" (Eckhart 2009, p. 505). Furthermore, "a man must always do one thing, he cannot do everything. It must always be one thing, and, in that one, one should take everything" (Eckhart 2009, p. 515). Whether it is a matter of multiple ways or one way, we should consider them together, as what Eckhart wants to emphasize is that this is the same thing. We should experience the omnipresence of God and understand that there is not just one path to God and that we cannot just worship God in the church and then forget about him outside the church. Nevertheless, one should also sway like a floating weed but trust in God, accept the path that God has given, and then go deep into it to realize God. That is the way, from one to all. A believer in God should not be obsessed with God but should experience God in his own life. He need not cultivate a love of God specifically but should realize that his love of something is always already the love of God, the way to God. "By observing whatever you are most inclined to or ready for: concentrate on that and observe yourself closely" (Eckhart 2009, p. 506). The same idea also appears in the neo-Vedānta of Vivekānanda and Aurobindo. The path to liberation is not only the wisdom yoga of Sankara. In Karma (act or work) Yoga, all works are seen as the path to liberation, wherein one must act without attachment to the results of the action.

Furthermore, we should not feel uneasy about the benefits of food, clothing, and shelter. Instead, we should accept them as they are, perceiving them as gifts from God. What is essential is not to be delighted by material things or saddened by personal loss. Following the same logic, even blasphemy seems to Eckhart to be praising God—blasphemy exerts the power of God, and this insulted God is not God but a creature. Everything is clearly one with God, and, when they work, "God works in them, not they themselves" (Eckhart 2009, p. 521). As both God and Divinity, He is the being that does all true works and also the nothing from where all works come; He demonstrates a synthesis of immanence and transcendence. Humans should act with detachment so that what is done is performed

by both humans and God. Correspondingly, Vedānta philosophy also proclaims that all works are God's work, and we should surrender the results of all works to God.

Considering the two kinds of union found in Catholic mystical theology, the statement here seems too much in favor of the essential union, emphasizing that God is omnipresent. This is because the mystical transforming union achieved by grace is already traditionally emphasized. Just as the philosophy of Vedānta is distinguished herein as Vivekānanda into four types of yoga, Karma Yoga is an "essential union", and Bhakti Yoga is a "mystical transforming union by Grace". Karma Yoga is a unique proposition from Vivekānanda and Bhakti Yoga is already present in the tradition. Although there are several ways of realizing "I am Brahman" in Indian Vedānta, here, we discuss only two ways of doing so. One is the tradition of non-dualism, represented by Śaṅkara, and the other is the neo-Vedānta (also called Hindu modernism), represented by Vivekānanda and Aurobindo.

As mentioned earlier, according to Śaṅkara's philosophy (Mādhavānanda 1921, pp. 45–46) of non-dualism, the biggest issue that humans face is Māyā, which means not clearly seeing that the essence of humans is Brahman. Therefore, to solve this problem, one needs to eliminate ignorance through discrimination, just as the mistaken idea of a snake is eliminated by discriminating between a snake and a rope. The specific practice involves renouncing identification with objective entities, focusing on the eternal Atman through meditation and rejecting the external universe and the world. Śaṅkara explained, "One should avoid the external objects and constantly apply oneself to meditation on the Atman. When the external world is shut out, the mind is cheerful … Hence the shutting out of the external world is the stepping-stone to Liberation" (Mādhavānanda 1921, pp. 147–48). Undoubtedly, Śaṅkara's advances in philosophy are profound. He completed a series of discourses after the Upaniṣads, premised on "I am Brahman", which represent some of his greatest contributions, namely, answering the question: "If I am Brahman, why is the appearance of man so different from that of Brahman?" However, the problem is that his method requires people to reject the external world and focus entirely on their inner selves. This generates a new difficulty: if Brahman is real, and Brahman is inherent in all things, then Brahman is also inherent in external things, so the external world should be real. Therefore, rejecting or closing off the external world and only focusing on the internal true self seems to make us lose a part of Brahman, a path to Brahman through the external world. The result, in India, is that practitioners focus on internal meditation, and more and more people do not participate in real social activities or else have a contemptuous attitude toward real life. This traditional thinking is a huge obstacle from the point of view of national and social development.

The religious reform movement that began in India in the early 19th century aimed to reform the traditional teachings of Vedānta, especially Śaṅkara's idea that the external world is an illusion. The most famous and influential figures of this movement were Swami Vivekānanda and Sri Aurobindo, who are considered representatives of the spirit of neo-Vedānta.

Vivekānanda's view of Māyā differs from Śaṅkara's, or rather, one could say that his attitude towards the external world is not as extreme as Śaṅkara's. Vivekānanda said that "Māyā is not a theory for the explanation of the world; it is simply a statement of facts as they exist, that the very basis of our being is contradiction…that wherever there is good, there must also be evil, and wherever there is evil, there must be some good, wherever there is life, death must follow as its shadow, and everyone who smiles will have to weep, and vice versa" (Vivekānanda 2019b). Like Śaṅkara, Vivekānanda believes that discriminating the real, Brahman, from the unreal can alleviate the suffering caused by contradiction. However, those who meditate are not greater than others who devote themselves to their duties and actions. Traditional Indian scriptures divide a person's life into four stages: "The Hindu begins life as a student; then he marries and becomes a householder; in old age he retires; and lastly he gives up the world and becomes a Sannyasin … No one of these stages is intrinsically superior to another" (Vivekānanda 2019a). Vivekānanda points out

that not only is every stage of life meaningful but also that all actions should be considered worship. The core of his Karma Yoga is self-sacrifice, advocating that people should dedicate the results of their actions to God without seeking the results or rewards of their actions. Here, we see another shift in Indian Vedānta philosophy, from inward meditation to the realizing of "I am Brahman" by various means—inward, secular, meditative, active, reclusive, devotional, and so on.

There is an interesting question: if Māyā refers to the obstacle that we now know we have to overcome in Vedānta, then what is the obstacle in Eckhart's philosophies? What lies between the human and the Divinity? Moreover, what of the things that we once believed should be a breakthrough, from Eckhart's viewpoint? One has to say that it is the personified God, who also represents the fact that Eckhart's philosophy had seen the body, the being, as an obstacle. Only because of the different standpoints of the two philosophies, Māyā is seen as an obstacle, and God is seen as what we break through.

There is a very close connection between Vivekānanda and Aurobindo, the latter being a true intellectual heir of the former. Aurobindo inherited the non-dualistic thought of Vedānta, but he also opposed the negation of the reality of the physical world and the severing of the link between Brahman and the physical world in non-dualism, which is one of the reasons why his philosophy is known as integral transformation. In short, Aurobindo was unsatisfied with the traditional Vedānta discussions of the relationship between Brahman, the self, and the world. He reconstructed the relationships between them, placing them within a continuous whole comprising Matter–Life–Mind–Spirit. In Aurobindo's view, unity with Brahman is not only to realize that one's innermost self is Brahman but also to realize that the gross physical body is also Brahman. His philosophy builds a gradual bridge between the seemingly opposite and incompatible matter and spirit. He believed that the Upaniṣad had already hinted at such a mystery: "Behind their (Matter and Spirit) appearances is the identity in essence of these two extreme terms of existence, in other words, 'Matter also is Brahman'" (Aurobindo 2005, p. 8). Between matter and Brahman, we can roughly discern life and mind. Nevertheless, they all exist within the One, as different intensities of force change continuously and cannot be isolated from each other. This model also appears in contemporary philosophy, such as with Bergson's concept of duration and Deleuze's philosophy of intensity. This model is like a spectrum, where we can see different ranges of color, but the substance that makes up each color of light is the same, and there are no strict boundaries between the ranges. We can only make vague distinctions. At the most subtle transition from red light to orange light, or the point where matter turns into life, i.e., the transition from inorganic to organic, we cannot separate them. In general, Aurobindo's Matter–Life–Mind–Spirit structure has the following characteristics: (a) Brahman pervades the entire structure, but "reality is not a sum or a concourse"; (b) there is a distinction between matter and substance. Substance can be regarded as Brahman, constituting matter, life, mind, and spirit (Aurobindo 2005, p. 38). We can talk about substance that constitutes matter, we can talk about purely dynamic life-energy substances, or we can construct the concept of entities, either philosophically or spiritually. Spirit itself is the pure substance of being, which is no longer the object of matter, life, or spiritual perception; however, it is the radiance of purely spiritual perceptual knowledge, in which the subject becomes its own object, that is to say, in which timelessness and spaceless are aware of themselves as the basis and primordial material of all existence in the self-extension of purely spiritual self-perception (Aurobindo 2005, p. 255). When the subject becomes its own object, it truly realizes that "I am Brahman"; (c) between matter and spirit, there is an infinite gradation (Aurobindo 2005, p. 268). As mentioned earlier, within this spectrum, there is no opposite or distinct entity.

Aurobindo (2005, pp. 8–19) strongly opposed the traditional non-dualistic view that the world is an illusion. He argued that since the world is also Brahman, there is no need to renounce the world to realize "I am Brahman". In his opinion, what is generally seen as being evil or obstructive does not exist as a concrete entity. It is merely the unreasonable demand among different ranges or forces. In the case of mind and life, for example,

when the need for matter exceeds a specific limit, matter becomes an obstructive force, but, at the same time, it is also the force by which matter maintains itself (Aurobindo 2005, pp. 258–59). This thought aligns with Eckhart's concept of compliance and the acceptance of everything that God gives. Thus, in Aurobindo's ashram, we see a pattern that is completely different from traditional Indian ashrams—no idols, no crosses, no monastic robes, no rituals, and no precepts. Practitioners must participate in productive labor and learn cultural knowledge; they must perform physical exercises and appreciate music and paintings. They must integrate their entire spiritual practice into every action in their daily life.

## 5. Conclusions

Eckhart's philosophy and sermons respond to the universal view and dualistic tradition that man and God are different from each other, responding to issue B: "Generally speaking, why is it that humans are not already one with God?" He tries to tell us that we must seek union with God by self-emptying. However, this union beyond time and space is so idealized and difficult that it has to respond to issue A, "How can we claim that humans (living in this finite world) are already one with God?" Here, humans no longer seek the personified God but rather experience the fundamental nothingness of the Divinity in detachment and, thus, recognize that everything expresses the Divinity. Fundamentally, Eckhart puts more emphasis on the Divinity and detachment. Vedānta begins with the idea that "I am Brahman"; at its starting point, this confronts issue A. In order to rationalize how a finite human can be Brahman, the concept of Māyā emerges. Just as Śaṅkara gives his view of the ego and of this world as unreal, a new difficulty arises: if there is something unreal in oneself, how can one say "I am Brahman"? Then Vedānta has to face issue B. Ultimately, Vedānta philosophy moves toward an emphasis on the concept that "all is Brahman", the idea that all work is Brahman itself acting with humans. This is the symmetrical structure within the development of Eckhart's and Vedānta philosophies, which culminates in their giving an essential role to the impersonal divine and emphasizing the importance of work and this worldly life.

In both Vedānta philosophy and Eckhart's works, God is considered perfect, omnipresent, and creating rather than being a creature. Thus, we have the Divine as nothing and the Brahman that which enfolds everything, which is also the relationship between the Divine and God in Eckhart's philosophy. As God is perfect, it seems to be inappropriate to attribute physical and mental pain and the imperfections of this worldly life to God, which gives rise to the idea of being one with God through the renunciation of the secular world and the body. Furthermore, God should be seen in all activities because God is omnipresent. A dilemma has arisen between issues A and B. This dilemma infuses Vedānta and Eckhart's philosophies with developments and tensions that bring out a kind of apparent contradiction in their different logical stages. These two philosophies each start from one pole of this dilemma and experience the other's issue at the other pole of the seesaw; both ultimately agree on the idea that all is One, the Divinity, indicating that all work is already the path to the One.

**Author Contributions:** Conceptualization, J.L. and Z.W.; methodology, J.L. and Z.W.; formal analysis, J.L.; writing—original draft preparation, J.L.; writing—review and editing, J.L.; supervision, Z.W. All authors have read and agreed to the published version of the manuscript.

**Funding:** This research received no external funding.

**Institutional Review Board Statement:** Not applicable.

**Informed Consent Statement:** Not applicable.

**Data Availability Statement:** No new data were created or analyzed in this study. Data sharing is not applicable to this article.

**Conflicts of Interest:** The authors declare no conflict of interest.

## Notes

[1] This tendency is evident in the fact that Eckhart consistently emphasizes the concepts of God and Divinity in various texts, aiming to guide individuals on how to seek and love God, as well as on how to lead a virtuous religious life. While Catholic mystical theology recognizes two forms of union—an essential union and a mystical transforming union achieved by grace— what appears to be crucial in Eckhart's philosophy is always the essential union, although the mystical transforming union has been important to earlier thinkers.

[2] "… delving deeper and ever seeking, she grasps God in His oneness and in his solitude … does not rest content but quests on to find out what it is that God is in His Godhead and in the ownness of His own nature" (Eckhart 2009, p. 338).

[3] According to Paul Deussen (1921, p. 21), while the ideas of the *Upaniṣads* originated among the Brahmins, they were developed and matured in the *kṣatriya* class before being accepted by the Brahmins. The ideas of the *Upaniṣads* did not initially receive strong support and promotion among the Brahmins. K.N. Upadhyaya (1938, p. 78) argued that Paul Deussen may not have considered the non-Brahmin origins of the ideas found in the *Upaniṣads*. However, their content was indeed influenced by the *kṣatriya* class, with many of the thinkers in the *Upaniṣads* belonging to this class. The point that I want to emphasize here is that while the *Upaniṣads* are generally considered to be part of the orthodox Brahmanical philosophy in India, their philosophical positions do conflict with the Vedas, especially in terms of the sacrificial rituals. In particular, the proposition "*I am Brahman*" could have been regarded as heresy by the Brahmin priests of the time.

[4] Eckhart gave examples such as: "Just as my eye cannot speak and my tongue cannot recognize colors, so love cannot incline to anything but goodness and God" (Eckhart 2009, p. 99).

[5] Vinzent (2011, pp. 57–58) demonstrated the rich origins of the word detachment.

[6] Vinzent (2011, pp. 223–26) explained in detail why detachment is superior to humility.

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
