# Peer review of "Balancing the Poles of the Seesaw: The Parallel Paths of Eckhart and Hindu Vedānta toward Oneness with God/Brahman"

_religions, doi:10.3390/rel14121529_

Round 1
Reviewer 1 Report
Comments and Suggestions for Authors
The topic of this paper is interesting, with writers from different traditions working on the same issue from different perspectives. But the sides need to be clarified. On one side we have the varieties of Vedanta, in the paper mostly Advaita Vedanta, which has maya as the barrier between the person and the divine. One the other side, we have Eckhart, where the barrier between the person and the divine is not as clear. Is it the body? The mind? Something else? What is the Western equivalent of maya?
In the conclusion, we have theistic Vedanta, and the similarity between Eckhart and Vedanta in terms of God. However, most of the paper deals with the non-theistic side, so this seems a bit stretched.
But the most important issue is why these distinctions and similarities are important. Why should the reader care about this? Are there some deeper concerns beyond it? What is the writer's opinion of universalism and cross-cultural religious truths? Is that the basis of the seesaw described by the writer?
Otherwise, my concerns would be minor. These sections are not full chapters, as they are described in line 50. There are some sentences that are unclear- should line 29 say 'unity' rather than 'unite'? Should line 35 be phrased more clearly, as 'this worldly life' rather than 'this world life?'
Otherwise, it would require some clarification on the position of Eckhart on the barrier between the person and God, and some emphasis on the significance of this for comparative world mysticism.
Comments on the Quality of English Language
There is too much jargon, but this is not an error, it is just a style that makes reading less pleasant than it could be.
Author Response
1.Comment: On one side we have the varieties of Vedantā, in the paper mostly Advaita Vedantā, which has Mayā as the barrier between the person and the divine. One the other side, we have Eckhart, where the barrier between the person and the divine is not as clear. Is it the body? The mind? Something else? What is the Western equivalent of Mayā?
1. Reply: Thank you for your comment. In Vedantā, Mayā is the barrier between humans and the divine; for Eckhart, due to the Christian backdrop, he naturally inherited the concept of humans being distinct from God. Both Vedantic philosophers and Eckhart grapple with the contrast of divine perfection and eternity versus human imperfection and temporality, encapsulated by Milne J.'s said, "the deep-seated notion that there are separate realities, namely, the differences between the universe and the Absolute, between human individuals, and between the Self or soul being distinct from God."
Vedantic philosophers attribute the duality to "Mayā"—an illusory force that makes us identify with impermanent objects as our true nature. Contrastingly, Eckhart, drawing from the Christian tradition, focuses on the inherent human-God difference while attempting to illustrate their sameness, thus not developing a mature concept like Mayā. However, in his philosophy, we can see his attempts to view the "self" as a barrier to our connection with Divinity, and even God itself also should be breakthrough. More significantly, in Vedantā or Eckhart's philosophy, the distinction between humans and God can be further split into the difference between humans and the personal God (Gott/Brahmā) and between humans and the divinity (Gottheit/Brahman). The article has added a discussion on this topic, and it will further highlight in the conclusion that the "personal God" is a barrier that must be transcended for the union of humans and "God". We have discussed this issue in lines 638-644.
2. Comment: In the conclusion, we have theistic Vedānta, and the similarity between Eckhart and Vedānta in terms of God. However, most of the paper deals with the non-theistic side, so this seems a bit stretched.
2. Reply: Thanks for the valuable feedback. The initial presentation in the manuscript was not clear enough. While most Vedantic scriptures refer to a personal God, Vedantic philosophy, which originates from the Upanishads, tends to lean more towards discussions of a non-personal God. In the revised version of the paper, we have incorporated a discussion differentiating between the Indian personal God (Brahmā) and the non-personal God (Brahman) in line 66, line 322. We hope these additions will provide greater clarity and depth to the manuscript, particularly in its treatment of non-theistic aspects of Eckhart's and Vedantic philosophies.
3. Comment: Why these distinctions and similarities are important? Why should the reader care about this? Are there some deeper concerns beyond it? What is the writer's opinion of universalism and cross-cultural religious truths? Is that the basis of the seesaw described by the writer?
3. Reply: Thanks for your questions. We believe these distinctions and similarities are critical because they reflect an ongoing transition within religious understanding, as impersonal aspects of God appear to be gaining prominence over personal elements. This shift correlates with advancements in science, technology, and human intellect.
In cross-cultural religious exchanges, personal gods often symbolise national identity. For instance, it may be challenging for a Chinese person to truly accept that the world evolved from Brahma in Hinduism, just as an Indian may find it challenging to embrace the idea that the universe was split open by Pangu, a figure in Chinese mythology.
However, what seems universally acceptable is the idea that the impersonal aspect of God, represented by the Hindu concept of "Brahman," the Chinese "Dao," and the Christian "God," is the same. These reflect our era's characteristics and indicate a future trend in religious development.
While different religious cultures experience unique stages and rhythms in the evolution from personal to impersonal aspects of God, this exploration appears to be a commonality in our development. Other articles, such as Milne J.'s "Eckhart and the problem of Christian non-dualism: A comparative study of Eckhart and Advaita Vedānta, have already discussed the similarities between these two philosophies.
As for universalism, despite different civilisations having different beginnings, the cultures of humanity have converged towards similar practices in their development. It represents an affirmation of globalisation and individual values.
I hope this response clarifies this research's importance and broader implications.
4. Comment: These sections are not full chapters, as they are described in line 50. There are some sentences that are unclear- should line 29 say 'unity' rather than 'unite'? Should line 35 be phrased more clearly, as 'this worldly life' rather than 'this world life?'
4. Reply: We appreciate your careful reading and constructive suggestions. We agree with your feedback and have made the recommended changes. The term in line 29 has been changed to 'unity,' and the phrase in line 35 has been revised to 'this worldly life.' We have also reviewed the rest of the manuscript and made corresponding adjustments where necessary.
5. Comment: It would require some clarification on the position of Eckhart on the barrier between the person and God, and some emphasis on the significance of this for comparative world mysticism.
5. Reply: Thank you for your insightful comment. We agree that Eckhart's position on the barrier between the person and God deserves more clarification. We have expanded the discussion to address these points in part 2 and part 4 of the paper. We hope this addition will provide a more comprehensive understanding.
Reviewer 2 Report
Comments and Suggestions for Authors
This article has the potential to become a very good article. However, there are still some things that need to be done. First of all, the comparison between Eckhart and Vedanta needs to be better theorized. The similarities will then come into better focus when their so different context is also better described. This is completely lacking. One sentence does refer to "several other papers," but there is no real conversation with scholarly debate. To become publishable, this really needs to change. A second problem is that there is very little reference to the texts themselves. It is mainly through secondary literature. The in itself original and intriguing thesis put forward by the author would become stronger if the starting point is taken in the texts.
Comments on the Quality of English Languageacceptable
Author Response
1. Comment: First of all, the comparison between Eckhart and Vedānta needs to be better theorized. The similarities will then come into better focus when their so different context is also better described. This is completely lacking. One sentence does refer to "several other papers," but there is no real conversation with scholarly debate. To become publishable, this really needs to change.
1. Reply: We appreciate your constructive feedback. In response to your suggestion, we have revised the Introduction section to include a more substantial engagement with the scholarly debate surrounding this topic. We have added studies like Milne J., Evola, and Shah-Kazemi in the first part, briefly explaining the key arguments presented in these studies, setting the context for the research in our article, and outlining the unique aspects of our article compared to these referenced works.
2. Comment: A second problem is that there is very little reference to the texts themselves. It is mainly through secondary literature. The in itself original and intriguing thesis put forward by the author would become stronger if the starting point is taken in the texts.
2. Reply: Thank you for your insightful feedback. We fully agree with you on this matter, though some compromises had to be made. In the section regarding Eckhart, we have relied on existing research to simplify our argument but have made an effort to use the English translation of Eckhart's work. Regarding the Vedānta section, while it may appear that we have predominantly cited secondary sources, we have drawn heavily from primary sources such as the Upanishads and Shankara's commentaries. These have been taken from English translations of the original Sanskrit, along with explanatory notes. When citing these sources, we indicated the information of the annotator, which might have given the impression of relying on secondary literature.
However, we acknowledge the importance of making this more explicit in the manuscript. We will revise the citations to reflect the use of primary sources and ensure that my reliance on these foundational texts is made more evident.
Reviewer 3 Report
Comments and Suggestions for Authors
As much as studies and research in comparative religion are laudable, it is to be presumed that the researcher embarking on such an endeavour is well read if not expert in the traditions being compared. In the case of this paper knowledge of the theological and philosophical subtleties is fundamental, especially in the area of mysticism where experiential knowledge is fundamental.
While the author shows a certain amount of reading, it seems to me that the paper does not succeed in giving an integrative perspective of Eckhart’s mysticism and Vedanta. With regards to Eckhart the author seems to bypass two major concepts:
1. The distinction between Gott (God) and Gottheit (Divinity). The first refers to a personal God of revelation and the second refers to what in Christian Theology it is called God’s essence. This is sheer unnameable simplicity. It seems to me, at least, that this distinction is unclear in the paper and as such it is a source of confusion on what is understood in the paper by “God.” Does the author have in mind Gott or Gottheit? Clarity on this matter is of utmost importance to convincingly develop the answer to the two questions of the paper: “How can we claim that humans are already one with God,” and “Why is it that humans are not already one with God.” (sic?). To my understanding he equivalent of Brahman is rather Gottheit, than Gott (see for example: Evola, Vedanta, Meister Eckhart and Shelling, 1960).
2. In Catholic Mystical Theology there is a distinction between essential union, referring to the divine Presence in everything, and the mystical transforming union by Grace. The paper seems to ignore this when trying to answer the research question. Failing to understand this one ends up misinterpreting Eckhart and totally goes out of point in establishing or pointing to parallelisms with Vedanta.
This confusion is clearly evident in the research question itself : “How can we claim that humans (living in this finite world) are already one with God,” and “Why is it that humans are not already one with God (so we are still living in this finite world).” (sic ?) - which keeps being repeated in the paper giving the impression that the author has no answer to it - and in statements like:
“he stresses the unity of our physical body with the body of God”;
“he needs to return to the middle point of the seesaw to maintain balance, which he achieves by pointing out that all is God, and all works are the works of God.”
The result of lack of clarity is an unconvincing section 4.
There are also some instances in reading the paper when one asks : “what about the principle of dissimilarity in Eckhart?”
The conclusion then seems more an opening to other arguments bringing forward other mystics and philosophers from the Islamic and Jewish worldview.
Comments on the Quality of English Language
English should be revised.
Author Response
1. Comment: The distinction between Gott (God) and Gottheit (Divinity). The first refers to a personal God of revelation and the second refers to what in Christian Theology it is called God’s essence. This is sheer unnameable simplicity. It seems to me, at least, that this distinction is unclear in the paper and as such it is a source of confusion on what is understood in the paper by “God.” Does the author have in mind Gott or Gottheit? Clarity on this matter is of utmost importance to convincingly develop the answer to the two questions of the paper: “How can we claim that humans are already one with God,” and “Why is it that humans are not already one with God.” (sic?). To my understanding he equivalent of Brahman is rather Gottheit, than Gott (see for example: Evola, Vedanta, Meister Eckhart and Shelling, 1960).
1. Reply: Thank you immensely. Your insight is indeed crucial to our revisions of this paper! We have reviewed the articles you mentioned and have extensively incorporated the distinction between God and Divinity throughout the article. Indeed, this has greatly enhanced the clarity of the article. The Introduction and Part 2 heavily increase the discussion of the distinction between God and divinity, Brahmā and Brahman. Corresponding improvements have been made throughout the relevant sections of the text, as you can see in the whole article. We have also summarised Evola’s article you mentioned in the Introduction.
2. Comment: In Catholic Mystical Theology there is a distinction between essential union, referring to the divine Presence in everything, and the mystical transforming union by Grace. The paper seems to ignore this when trying to answer the research question. Failing to understand this one ends up misinterpreting Eckhart and totally goes out of point in establishing or pointing to parallelisms with Vedanta.
This confusion is clearly evident in the research question itself: “How can we claim that humans (living in this finite world) are already one with God,” and “Why is it that humans are not already one with God (so we are still living in this finite world).” (sic ?) - which keeps being repeated in the paper giving the impression that the author has no answer to it - and in statements like:
“he stresses the unity of our physical body with the body of God”; “he needs to return to the middle point of the seesaw to maintain balance, which he achieves by pointing out that all is God, and all works are the works of God.”
The result of lack of clarity is an unconvincing section 4.
2. Reply: Thank you for your insightful comments. In our view, although Eckhart's philosophy emphasizes both types of union, the latter – the mystical transforming union by Grace – has been crucial in previous traditional thought. Eckhart's theory of "essential union" highlights his uniqueness. Eckhart always discusses God and divinity in terms of how individuals can seek God, love God, and live a good religious life. This makes the "essential union" even more significant. Much like the distinction in Vedānta philosophy as explained by Vivekananda, which is divided into four kinds of yoga: Jnana Yoga, Bhakti Yoga, Karma Yoga and Raja Yoga. Karma Yoga can be viewed as an "essential union," while Bhakti Yoga represents a "mystical transforming union by Grace." Karma Yoga is a unique proposition from Vivekananda, while Bhakti Yoga already exists in tradition.
Of course, we are not experts in Catholic Mystical Theology, and our grasp of the overarching concept may not be accurate. We sincerely welcome your criticism and guidance if there are errors in our understanding. For now, we have added a footnote in the Introduction (line 82) to clarify the bias of this article, and we have stressed the distinction between the two types of union in Chapter 4 (lines 581-587).
3. Comment: There are also some instances in reading the paper when one asks: “what about the principle of dissimilarity in Eckhart?” The conclusion then seems more an opening to other arguments bringing forward other mystics and philosophers from the Islamic and Jewish worldview.
3. Reply: Thank you for highlighting the critical principle of dissimilarity in Eckhart. We recognize the importance of the principle of dissimilarity, which is chiefly reflected in this paper's emphasis on the difference between the human and the divine in the so-called Eckhartian thought. As stated in this paper, the progression from the difference between the human and the divine to their unity is not logically presented in Eckhart's philosophy. It isn't a case of one stage replacing the previous one; all these stages coexist, embodying Eckhart's thought's inherent tension and richness. Dissimilarity is also a part of it.
You are right in noting that our paper should have conceptually stressed this point. Taking your feedback into account, we have emphasised the essence of creatures is nothingness in the text (line 161), pointed out dissimilarity in the Introduction with the help of Schürmann's description (line 94), and further emphasised the coexistence of stages in Eckhart's philosophy ( line 89). In addition, we have a reference to God (Divinity) always beyond the creature at the end of part 2 (line 272)
We have essentially rewritten the conclusion, focusing on the main topic of discussion in this paper, namely the symmetry in the structure of Vedānta and Eckhart's philosophies. We hope this rewrite is more precise and detailed than the original manuscript.
Round 2
Reviewer 3 Report
Comments and Suggestions for Authors
Thankyou for major improvement and more clarity.
Comments on the Quality of English LanguageCheck english for minor revisions.